# SPARSE POLICY SPACE RESPONSE ORACLES

## ABSTRACT

In multi-agent non-transitive games, the Policy Space Response Oracles (PSRO) framework approximates Nash Equilibrium by iteratively expanding policy populations. However, the framework suffers from severe policy redundancy in the processes of policy generation and policy population construction, thereby leading to a substantial increase in computational complexity. To address these limitations, this paper proposes Sparse PSRO, a novel framework that overcomes policy redundancy through two key innovations: (1) Sparsity Metric, which quantifies the dissimilarity between candidate strategies and existing populations via convex combination residual constraints, guiding the algorithm to explore underrepresented payoff regions while suppressing redundant policy generation; (2) Policy Space Sparsification, which constructs the Policy Hull backbone via intensive early exploration and admits only geometrically distinct strategies through threshold control, effectively reducing the number of policies and lowering computational complexity. Theoretical analysis proves that Sparse PSRO maintains a finite policy population with guaranteed separation distances, preventing exponential population growth while ensuring convergence to the Nash Equilibrium. Experiments across diverse environments (including RGoS, AlphaStar888, Blotto, and Kuhn Poker) demonstrate that Sparse PSRO significantly outperforms six baseline methods in terms of exploitability and policy population size, thus validating its effectiveness in efficiently approximating Nash Equilibrium with reduced computational costs.

## 1 INTRODUCTION

In the field of multi-agent reinforcement learning, two-player zero-sum games have become a core focus of theoretical and applied research due to their analytical tractability and symmetry. The Nash Equilibrium (NE) (Nash Jr, 1950), a central solution concept in game theory, ensures optimal performance in worst-case scenarios. Traditional equilibrium computation methods aim to identify these solutions, such as by enumerating all equilibria (e.g., the work of Avis et al. (2010) for bimatrix games) or finding a single sample equilibrium (e.g., the Lemke-Howson algorithm for bimatrix games or linear programming for zero-sum matrix games (Von Stengel, 2002)).

However, in complex real-world environments characterized by strong non-transitivity (Czarnecki et al., 2020), as exemplified by strategic games like StarCraft (Vinyals et al., 2019) and poker, traditional game-theoretic algorithms frequently fail to converge to Nash Equilibrium due to inadequate exploration of the exponential policy space (Lanctot et al., 2017). This fundamental challenge stems from the cyclical dominance patterns inherent in non-transitive systems (illustrated by the canonical Rock-Paper-Scissors paradigm) (Candogan et al., 2011; Balduzzi et al., 2018). Such non-transitivity is a pervasive phenomenon across a wide range of games (Sanjaya et al., 2022; Li et al., 2023). The Policy Space Response Oracle (PSRO) (Lanctot et al., 2017) framework, which approximates NE by iteratively expanding policy populations, has emerged as the mainstream approach for handling non-transitive games (Bighashdel et al., 2024). Its core logic relies on a cycle of "meta-game solving-best response generation-population update", leveraging policy diversity to drive algorithmic convergence. Nevertheless, traditional PSRO faces critical challenges in practice: policy redundancy during both policy generation and policy population construction processes leads to low exploration efficiency (Liu et al., 2022b), and the accumulation of redundant policies sharply increases computational costs, thereby limiting its practical applicability in complex games.

Existing studies have proposed various improvements to address PSRO's limitations. In terms of reducing policy redundancy during policy generation, some works enhance policy diversity through

diversity regularization, such as using determinantal point processes (DPP) (Perez-Nieves et al., 2021) to quantify the geometric diversity of policy distributions or integrating multi-dimensional metrics via the Unified Diversity Measure (UDM)(Liu et al., 2022b). However, these methods ignore the essential disparities in policy payoff patterns and fail to establish a direct theoretical link between diversity and equilibrium quality. To address these issues, Policy Space Diversity PSRO (PSD-PSRO) (Yao et al., 2023) proposes the concept of Population Exploitability (PE) to measure the strength of a population. It optimizes the geometric coverage of policy distributions via KL divergence. Yet PSD-PSRO adopts an approximation based on minimum KL divergence for policy generation, which may not fully guarantee Policy Hull expansion. In terms of reducing policy redundancy during policy population construction, the extent of exploration in existing studies remains notably limited. Additionally, there are studies that improve computational efficiency through distributed training, such as parallelized PSRO (McAleer et al., 2020). However, these methods overlook the issue of policy redundancy, thereby limiting the efficiency of policy space expansion.

To address these challenges, we propose Sparse PSRO, a novel framework that enables efficient NE calculation through two components: (1) Sparsity Metric: which employs a convex combination residual metric to quantify policy dissimilarity, thus eliminating policy redundancy during the policy generation process. (2) Policy Space Sparsification: which applies threshold-controlled updates to admit only geometrically distinct strategies, thus addressing policy redundancy during policy population construction. Furthermore, we provide theoretical proofs that Sparse PSRO maintains a finite policy population with guaranteed separation distances and converges to NE. Empirically, we compare Sparse PSRO against multiple baselines across four tasks, including RGoS, AlphaStar888, Blotto, and Kuhn Poker. The results demonstrate that Sparse PSRO significantly outperforms baselines in terms of exploitability and policy population size, validating its effectiveness. Our primary contributions are as follows:

- We pinpoint that addressing policy redundancy during both policy generation and policy population construction is critical for enhancing PSRO's computational efficiency when approximating NE in two-player symmetric zero-sum games.

- Accordingly, we propose Sparse PSRO, which employs Sparse Metric and Policy Space Sparsification to address the policy redundancy issue, thus enabling efficient NE calculation.

- Theoretical analysis proves that Sparse PSRO maintains a finite policy population with guaranteed separation distances and convergence to Nash Equilibrium. Furthermore, extensive experiments demonstrate that Sparse PSRO outperforms six baseline methods in exploitability and policy population size.

## 2 RELATED WORK

As mentioned above, we pinpoint that PSRO suffers from policy redundancy during both policy generation and policy population construction. Accordingly, we divide existing works into two major categories: policy generation (focusing on diversifying strategy creation) and policy population construction (optimizing population management).

**Policy Generation:** Policy generation research aims to produce diverse and high-quality strategies through structured optimization and regularization. From the perspectives of game theory, many studies aim to improve policy diversity by expanding the Gamescape (Balduzzi et al., 2019)(the convex hull of policy payoff vectors). PSRO-rN (Balduzzi et al., 2019) introduces rectified Nash response, which selectively expands the Gamescape by amplifying policy advantages. Theoretical proofs show it strictly enhances policy response capabilities. BD&RD-PSRO (Liu et al., 2021), integrating behavioral and response diversity, provides a multi-dimensional evaluation framework. DPP-PSRO (Perez-Nieves et al., 2021) selects policies by maximizing expected cardinality to avoid generating redundant policies. PSD-PSRO (Yao et al., 2023) further proposes the concept of Policy Hull and optimizes the geometric coverage of policy distributions via KL divergence. Pipeline PSRO (McAleer et al., 2020) accelerates convergence through hierarchical pipeline parallel training, demonstrating significant advantages in large-scale games. Notably, there is an intrinsic link between *efficiency optimization* and *diversity enhancement*—algorithms with fast convergence often require precise policy selection mechanisms, providing important insights for subsequent sparsity research.

**Policy Population Construction:** To control population growth, threshold-based admission mechanisms were developed. A notable complementary approach is the threshold-controlled strategy addition introduced in A-PSRO (Hu et al., 2023), which admits new strategies into the population only when their payoff improvement exceeds a predefined threshold. This sparsification not only reduces computational requirements but also enhances learning efficiency, making the algorithm more advantageous in handling large-scale data.

Notably, existing works fail to simultaneously address policy redundancy in both policy generation and policy population construction. To address this issue, we propose Sparse PSRO, which unifies sparsity-aware generation (alleviating the policy redundancy during policy generation) with geometrically controlled population construction (addressing the policy redundancy during policy population construction) to address both dimensions simultaneously. These mechanisms collectively enable efficient Nash Equilibrium calculation in two-player zero-sum games.

## 3 PRELIMINARIES

### 3.1 SYMMETRIC ZERO-SUM GAMES, NE, AND EXPLOITABILITY

**Symmetric zero-sum games** can be rigorously formalized as $G = \{S, U\}$. In this game structure, both players share the policy space $S$, and the payoff matrix $U$ satisfies the property $U = -U^\top$, indicating that the payoff obtained by one party is exactly equal to the loss suffered by the other. Regarding the types of strategies, a pure strategy $\pi \in S$ corresponds to a deterministic action choice made by the player, while a mixed strategy $\pi_i \in \Delta(S)$ is a probability distribution, which can be represented as a weighted combination of pure strategies.. For two players $i \in \{1, 2\}$, its payoff is precisely measured by the function $u(\pi_1, \pi_2) = \pi_1 \cdot U \cdot \pi_2^\top$.

**Nash Equilibrium**, as a core solution concept in game theory, holds crucial significance. When the strategy profile $(\pi_1^*, \pi_2^*)$ satisfies $\forall P_i, u(\pi_1^*, \pi_2^*) \geq u(\pi_1, \pi_2^*)$ and $u(\pi_1^*, \pi_2^*) \geq u(\pi_1^*, \pi_2)$ for any player $i$, the strategy profile reaches the Nash Equilibrium state. In this state, neither player can increase their payoff by unilaterally adjusting their own strategy. The Best Response (BR) of player $i$ to the opponent's strategy $\pi_{-i}$ is denoted by:

$$\mathcal{BR}(\pi_{-i}) = \arg\max_{\pi_i'} u(\pi_i', \pi_{-i}) \tag{1}$$

In Nash Equilibrium, each player's strategy constitutes a best response to that of the other.

To accurately quantify the deviation of a strategy profile from the Nash Equilibrium, Lanctot et al. (2017) introduces the important concept of **Exploitability**. Its mathematical definition is as follows:

$$\mathcal{E}(\pi) = \frac{1}{2} \sum_{i \in \{1,2\}} \left( \max_{\pi_i'} u(\pi_i', \pi_{-i}) - u(\pi_i, \pi_{-i}) \right) \tag{2}$$

When $\mathcal{E}(\pi) = 0$, it clearly indicates that the strategy profile $\pi$ has reached the Nash Equilibrium state.

To quantify the equilibrium approximation quality of policy populations (sets of strategies accumulated over iterations, denoted $\Pi_i^t = \{\pi_i^1, \ldots, \pi_i^t\}$ for player $i$ at iteration $t$), two key concepts form the theoretical foundation. The **Policy Hull** (Yao et al., 2023) is defined as the set of all mixed policies formed by convex combinations of policies in the population:

$$H(\Pi_i) = \{\sum_j \beta_j \pi_i^j \mid \beta_j \geq 0, \sum_j \beta_j = 1\} \tag{3}$$

The volume of the Policy Hull intuitively reflects the richness of the policy space.

**Population Exploitability (PE)** (Yao et al., 2023) measures the deviation of the joint policy population $\Pi = \Pi_i \times \Pi_{-i}$ from a global NE:

$$\mathcal{PE}(\Pi) = \frac{1}{2} \sum_{i \in \{1,2\}} \max_{\Pi_i' \subseteq \Omega_i} \mathcal{P}_i (\Pi_i', \Pi_{-i}) \tag{4}$$

where $\mathcal{P}_i$ denotes relative population performance, and $\Omega_i$ is the full policy space of player $i$. The summary of notations is available in Appendix A.

## 3.2 Policy Space Response Oracle Framework

When solving the problem of approximate Nash Equilibrium in non-transitive games, the Policy Space Response Oracle has become the current mainstream framework choice. The core operation mechanism of this framework lies in continuously iteratively expanding the policy population to steadily approach the Nash Equilibrium. PSRO consists of the following three related key steps:

**Meta-game Construction:** For each player $i$, PSRO maintains a policy population $\Pi_i^t = \{\pi_i^1, \ldots, \pi_i^t\}$, which forms the basis of the policy space for the meta-game. In the meta-game, the payoff matrix $M_{\Pi_i, \Pi_{-i}}$ records in detail the payoffs generated by the interactions between different policies, where $M[j, k] = u(\pi_i^j, \pi_{-i}^k)$.

**Meta-policy Solving:** By solving the meta-game, the Nash Equilibrium $\sigma_i \in \Delta(\Pi_i^t)$ is obtained. This result represents the optimal mixed policy under the current policy population and precisely characterizes the player's probabilistic choice tendency at the policy level.

**Best Response Generation:** In response to the $-i$'s meta-policy $\sigma_{-i}^t$ at iteration $t$, the player generates a new policy $\pi_i^{t+1} \in \mathcal{BR}(\sigma_{-i}^t)$ and incorporates it into the policy population, thereby achieving the gradual expansion of the policy space.

Although the PSRO framework uses the cycle of "solving meta-equilibrium-generating best response-updating the population" and leverages the diversity of policies to drive the algorithm towards the Nash Equilibrium, traditional PSRO methods are extremely prone to the problem of policy redundancy in practical applications (Liu et al., 2021). This leads to a sharp increase in computational costs, which limits its effective application in large-scale game scenarios to a certain extent.

# 4 Sparse PSRO

We begin by introducing a sparsity metric for policy diversity, which eliminates policy redundancy during the policy generation process. We then delve into the process of achieving Policy Space Sparsification, which enables the removal of policy redundancy during policy population construction. Subsequently, we provide theoretical validation that Sprse PSRO maintains a finite policy population with guaranteed separation distances and convergence to the Nash Equilibrium. Finally, we summarize the overall learning procedure of Sparse PSRO.

## 4.1 Policy generation

In multi-agent game scenarios, the computational complexity of solving Nash Equilibria grows exponentially with the dimensionality of the policy space. According to Carathéodory's theorem (Althöfer, 1994), any Nash Equilibrium solution in a finite game can be expressed as a convex combination of a finite set of pure strategies. Formally, each player $i$'s mixed strategy NE can be represented as follows:

$$\sigma_i = \sum_{k=1}^{m} \alpha_k \pi^k, \quad \alpha_k \geq 0, \sum_{k=1}^{m} \alpha_k = 1 \tag{5}$$

where $\pi^k$ denotes the $k$-th pure strategy, $m$ denotes the number of pure strategies, and $\alpha_k$ represents the mixing probability for the corresponding pure strategy. This property highlights that Nash Equilibrium solutions are inherently convex combinations of existing pure strategies.

For a candidate strategy $\pi_i$, Sparse PSRO defines the sparsity metric as its convex combination residual relative to the $t$-th iteration policy population $\Pi_i^t$:

$$\mathrm{sparsity}(\pi_i, \Pi_i^t) = \min_{\substack{1^\top \mathbf{c} = 1 \\ \mathbf{c} \geq 0}} \|\pi_i - {\Pi_i^t}^\top \mathbf{c}\|_2^2 \tag{6}$$

where $\mathbf{c} = [c_1, c_2, \ldots, c_m]^\top$ is a vector of convex combination coefficients. This formula quantifies the *global difference* between the new strategy and existing strategies by minimizing the Euclidean distance between the new strategy and the convex combination of existing strategies.

Policy Hull is the convex combination of policies in the population. Theoretically, expanding the Policy Hull has been proven to reduce population exploitability, thereby accelerating the algorithm's convergence to the Nash Equilibrium (Yao et al., 2023). Specifically, if newly generated policies can significantly enlarge the Policy Hull, the algorithm can more efficiently cover potential equilibrium policies and improve convergence efficiency. A larger residual indicates that the new strategy is more likely to lie outside the current Policy Hull, thereby effectively promoting Policy Hull expansion to reduce Population Exploitability.

Building on this foundation, Sparse PSRO incorporates the policy sparsity as a regularization term into the best-response objective function, forming an improved optimization formulation:

$$\pi_i^{t+1} = \arg\max_{\pi_i} \left[ u(\pi_i, \sigma_{-i}^t) + \lambda \cdot \text{sparsity}(\pi_i, \Pi_i^t) \right] \tag{7}$$

where $\sigma_{-i}^t$ denotes the opponent's meta-policy at iteration $t$, and $\lambda > 0$ controls the weight of the sparsity metric in the best-response optimization process. By explicitly constraining the convex combination residual of the new strategy relative to the existing policy population, this formulation drives the algorithm to enable more diverse policies. Strategies with large convex combination residuals are more likely to cover regions that remain unexplored by the current Policy Hull.

## 4.2 POLICY SPACE SPARSIFICATION

To address the policy redundancy in policy population construction, Sparse PSRO employs a threshold-controlled strategy addition rule that controls the Policy Space Sparsification of the policy population $\Pi_i$, the policy population is updated as:

$$\Pi_i^{t+1} = \begin{cases} \Pi_i^t, & \text{if sparsity}(\pi_i, \Pi_i^t) \leq \mu \\ \Pi_i^t \cup \{\pi_i\}, & \text{otherwise} \end{cases} \tag{8}$$

where $\mu > 0$ is a predefined threshold. This rule ensures that only strategies sufficiently distant from the existing Policy Hull are added, preventing redundant inclusion of similar strategies.

This mechanism allows Sparse PSRO to quickly build the policy space's foundational structure and maintain diversity through Threshold-Controlled additions, mitigating training instability from excessively low-quality strategies and ensuring stable convergence. The threshold-controlled strategy addition mechanism not only ensures computational efficiency but also provides theoretical guarantees for population finiteness, formally stated as follows:

**Theorem 4.1.** *Assume the policy space $\mathcal{S}$ is a compact subset of a finite-dimensional normed vector space, and let $\mu > 0$ be the sparsity threshold in Eq. 8. For any initial policy population $\Pi_i^1$, the policy population $\Pi_i^t$ maintained by Sparse PSRO remains finite for all $t \geq 1$.*

The proof is provided in Appendix B.1. Theorem 4.1 guarantees that the policy population remains tractable, avoiding exponential growth while preserving sufficient diversity for equilibrium approximation. The finite policy population ensures that the meta-game payoff matrix $M_{\Pi_i, \Pi_{-i}}$ remains of bounded size, lowering the computational complexity of meta-game solving across iterations.

## 4.3 OVERALL LEARNING PROCEDURE

The Sparse PSRO algorithm integrates Sparsity Metric and Policy Space Sparsification into the PSRO framework, enabling efficient policy exploration while maintaining computational tractability. The algorithm alternates between solving the meta-game for current policies, generating new best responses with sparsity regularization, and updating the policy population under threshold control.

Unlike traditional PSRO, which may accumulate redundant strategies and suffer from exponential population growth, Sparse PSRO's policy space sparsification ensures that only geometrically distinct strategies are retained. This not only *reduces computational costs* but also guarantees that the Policy Hull *expands efficiently*, thereby maintaining convergence stability.

We provide the pseudo-code of Sparse PSRO in Algorithm 1. UNIFORM denotes random sampling according to the uniform distribution. Sparse PSRO calculates the payoff matrix and initializes meta-

---

**Algorithm 1** Sparse PSRO

---

**Input** Initial policy populations $\Pi_i^1, \Pi_{-i}^1$
 1: Compute payoff matrix $M_{\Pi_i^1, \Pi_{-i}^1}$
 2: Initialize meta policies $\sigma_i^1 \sim \text{UNIFORM}(\Pi_i^1)$          ▷ Initialize meta-policies with uniform distribution
 3: **for** $t = 1, 2, \ldots$ **do**
 4:      **for** player $i \in \{1, 2\}$ **do**
 5:          Initialize $\pi_i = \pi_i^t$          ▷ Load current policy as optimization starting point
 6:          Sample $K$ policies $\{\pi_i^k\}_{k=1}^K$ from Policy Hull $\Pi_i^t$    ▷ Sample policies from Policy Hull
 7:          **for** many episodes **do**
 8:              Sample $\pi_{-i} \sim \sigma_{-i}^t$       ▷ Sample opponent policy from meta-policy distribution
 9:              Update $\pi_i$ over Eq. 7       ▷ Apply sparsity-regularized best-response optimization
10:          **end for**
11:          $\pi_i^{t+1} = \pi_i$
12:          **if** $\text{sparsity}(\pi_i, \Pi_i^t) > \mu$ **then**          ▷ Threshold-controlled policy admission
13:              $\Pi_i^{t+1} = \Pi_i^t \cup \{\pi_i\}$       ▷ Admit only geometrically distinct policies to population
14:          **end if**
15:      **end for**
16:      Compute missing entries in $M_{\Pi_i^t, \Pi_{-i}^t}$
17:      Compute meta-strategies $(\sigma_i^{t+1}, \sigma_{-i}^{t+1})$ from $M_{\Pi_i^t, \Pi_{-i}^t}$
18: **end for**
**Output** current meta-strategy for each player.

---

policies via uniform sampling during the initialization phase (Lines 1-3), then updates the player's policy through sparsity-regularized best-response in the policy generation phase (Lines 6-10). The core sparsification mechanism adds the new policy to the population only when its sparsity exceeds the threshold $\mu$ during the policy population construction phase (Lines 12-13), while the meta-game update phase fills in missing entries of the payoff matrix and computes new meta-strategies for the next iteration (Lines 16-17).

As the core parameter for policy selection, the rationality of the threshold $\mu$ directly determines whether the framework can retain strategies critical to Policy Hull expansion, thereby influencing the final convergence effect. To lay the necessary foundation for subsequently proving the convergence of Sparse PSRO, it is first necessary to clarify the key setting conditions of the threshold $\mu$, as specified in the following assumption:

**Assumption 4.2.** The threshold $\mu$ is sufficiently small to retain strategies critical for expanding Policy Hull, ensuring that strategies with the potential to expand the Policy Hull are not eliminated.

Based on the aforementioned assumption regarding the threshold $\mu$, we can further verify the convergence of Sparse PSRO to the global Nash Equilibrium from a theoretical perspective, as well as the relationship between Policy Hull expansion and the reduction of Population Exploitability. The specific conclusions are proven in the following theorem:

**Theorem 4.3.** *Sparse PSRO converges to a global NE of the full game. As long as the Population Exploitability of the joint policy population remains positive, adding the optimal strategy generated by Sparse-PSRO's update rule strictly expands the Policy Hull and reduces Population Exploitability.*

The proof of Theorem 4.3 is in Appendix B.2.

## 4.4 DISCUSSION

Sparse PSRO imposes structured constraints on the strategy update process through an approximate linear representation framework, ensuring new strategies are geometrically distinct from the existing policy population. This design fundamentally differentiates it from PSD-PSRO (Yao et al., 2023), which relies on KL divergence to measure probabilistic distribution differences without explicit exploitation of policy structural information. We emphasize the difference between them below.

PSD-PSRO has a critical theoretical inconsistency. Although its theoretical framework relies on the expansion of the Policy Hull to reduce Population Exploitability, its actual implementation adopts an approximate approach (minimum KL divergence from the strategy to the population's vertex strategies), resulting in a disconnect between theoretical guarantees and engineering implementation. In contrast, Sparse PSRO achieves an effect equivalent to Policy Hull expansion by employing a constrained sparsity metric while maintaining *intrinsic consistency* with the representation of Nash Equilibrium, thereby *resolving this contradiction* between theory and implementation.

The proposed sparsity metric measures the distance between a newly generated policy and existing strategies from a *global policy population perspective*. This method overcomes the locality limitation of the pairwise comparisons in the KL divergence and more accurately identifies strategies that can significantly expand the Policy Hull.

## 5 EXPERIMENT

In this section, we design experiments to answer the following questions: (1) Can Sparse PSRO achieve efficient Nash Equilibrium approximation by reducing policy redundancy while maintaining low exploitability? (See Sec. 5.1) (2) Which component contributes most to its performance gains: Sparsity Metric or Policy Space Sparsification? (See Sec. 5.2)

For question (1), we compare our method against state-of-the-art PSRO variants covering key research directions such as diversity regularization, computational efficiency, and theoretical convergence enhancement. These methods include PSRO (Lanctot et al., 2017), Pipeline PSRO (McAleer et al., 2020), $PSRO_{rN}$ (Balduzzi et al., 2019), DPP-PSRO (Perez-Nieves et al., 2021), BD&RD-PSRO (Liu et al., 2021), and PSD-PSRO (Yao et al., 2023). These methods form a multi-dimensional benchmarking system to ensure comprehensive and targeted comparisons. We use two metrics to quantify performance: exploitability (Eq. (2)) and population exploitability (Eq. (4)), which measure how much a policy population deviates from the NE. Additionally, we monitor policy population size to assess computational efficiency, as redundant policies directly impact training costs.

For question (2), we conduct ablation studies to evaluate the effectiveness of each key component in Sparse PSRO. To validate the necessity of core innovative modules, we design two ablation variants:

- Sparse PSRO without Sparsity Metric(Sparsification-PSRO): Removes the convex combination residual constraint, retaining only the best-response optimization objective to isolate the effect of sparsity regularization.
- Sparse PSRO without Policy Space Sparsification(Sparsity-PSRO): Replaces Policy Space Sparsification with a continuous dense exploration mode to verify the role of the threshold-controlled strategy addition rule in exploration-exploitation balance.

Experiments are conducted across typical Real-World games (Czarnecki et al., 2020): Random Game of Skills (RGoS), AlphaStar, Blotto, and Kuhn Poker. Ablation variant algorithms and additional experimental details are provided in Appendices C and D.

### 5.1 COMPARATIVE RESULTS

In terms of exploitability, as shown in Fig. 1, Sparse PSRO achieves the lowest exploitability in all environments, indicating a closer approximation to the Nash Equilibrium. In complex games like AlphaStar888, it maintains lower exploitability throughout iterations, outperforming baseline methods such as PSD-PSRO and BD&RD-PSRO. In classic non-transitive games (e.g., Kuhn Poker and Blotto), Sparse PSRO also exhibits faster convergence to low exploitability, validating its scalability.

In terms of population exploitability, the results in Fig. 2 further confirm this advantage: Sparse PSRO's PE declines more rapidly and stabilizes at a lower level compared to alternatives. This suggests that the policy population's policy hull expands more effectively, enabling the population to cover equilibrium strategies with higher efficiency.

In terms of computational efficiency, Sparse PSRO's policy population size remains significantly smaller than most baselines. As shown in Tab. 1, which presents policy population sizes across environments, our Sparse-PSRO maintains a compact population in all tested scenarios. For instance,

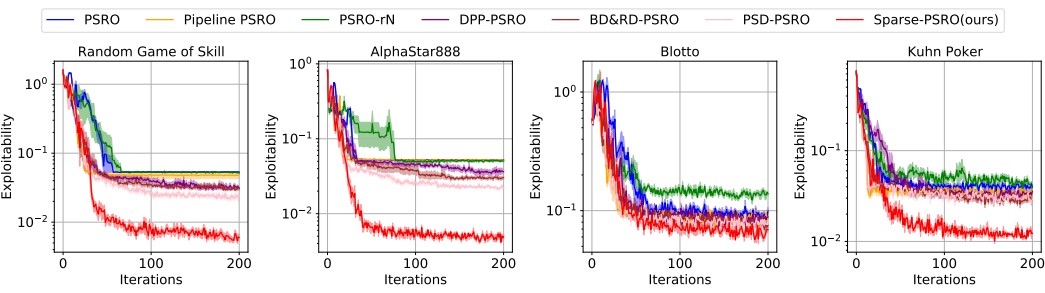

Figure 1: Comparison of exploitability across environments

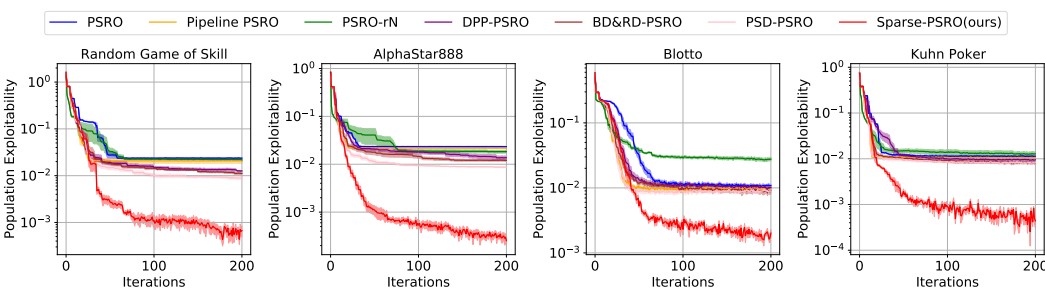

Figure 2: Comparison of population exploitability across environments

in Blotto and Kuhn Poker, it avoids the exponential growth seen in Pipeline PSRO and PSRO$_{rN}$, reducing the complexity of meta-game solving while preserving policy diversity. This balance between exploration quality and computational cost is a key advantage of the proposed framework.

Table 1: Policy population sizes across environments

| Algorithm | Random Game of Skills | AlphaStar888 | Blotto | Kuhn Poker |
|---|---|---|---|---|
| PSRO | 42 | 51 | 67 | 66 |
| Pipeline PSRO | 82 | 101 | 131 | 134 |
| PSRO$_{rN}$ | 82 | 111 | 187 | 179 |
| DPP-PSRO | 80 | 95 | 118 | 122 |
| BD&RD-PSRO | 77 | 91 | 117 | 119 |
| PSD-PSRO | 80 | 99 | 135 | 134 |
| **Sparsity-PSRO (ours)** | 86 | 104 | 133 | 132 |
| **Sparsification-PSRO (ours)** | 69 | 70 | 91 | 73 |
| **Sparse-PSRO (ours)** | 69 | 70 | 90 | 73 |

## 5.2 ABLATION RESULTS

As shown in Fig. 3, Sparse PSRO consistently achieves the lowest exploitability in all environments. In contrast, the ablation variants exhibit marked performance degradation: Sparsification-PSRO and Sparsity-PSRO both show higher exploitability values, with gaps widening over iterations. This indicates that omitting either component impairs the algorithm's ability to approximate the Nash Equilibrium. Specifically, when the Sparsity Metric is removed, the resulting policies lack geometric diversity, leading to insufficient coverage of the policy space. When Policy Space Sparsification is removed, redundant strategies accumulate, thereby diluting the effectiveness of the policy population.

Fig. 4 further validates this pattern by showing that Sparse-PSRO maintains the lowest PE across all environments, reflecting more efficient expansion of the policy hull. The ablation variants, however,

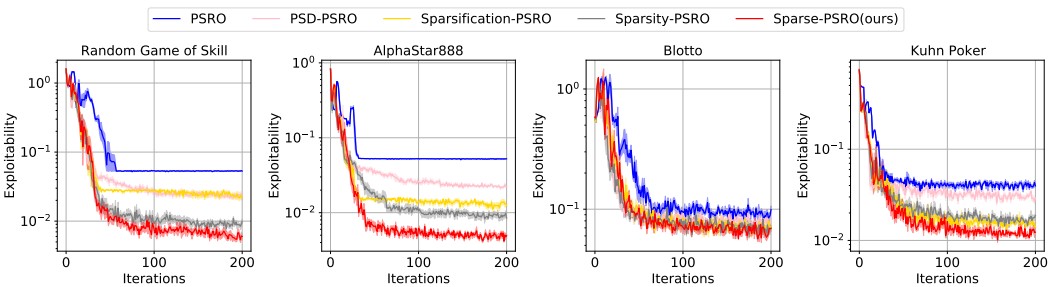

Figure 3: Ablation study on exploitability across environments

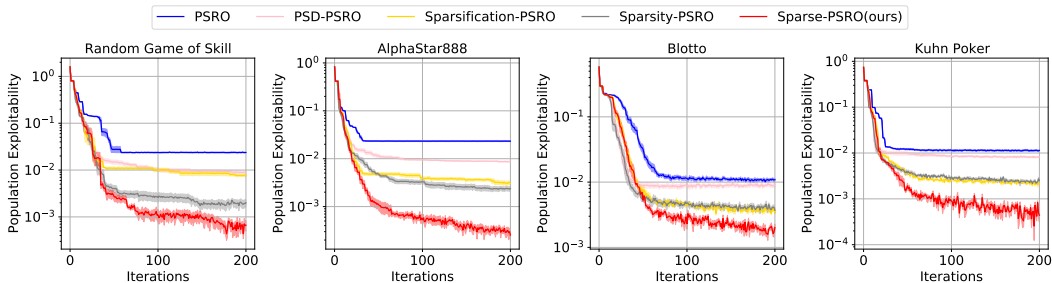

Figure 4: Ablation study on population exploitability across environments

exhibit slower PE decline and stabilize at higher values, confirming that the synergy between the two components is essential for reducing the population's vulnerability to exploitation.

The policy population size data in Tab. 1 further validates the role of Policy Space Sparsification. Sparsity-PSRO has a significantly larger population size compared to Sparse PSRO, approaching the size of baseline methods like PSD-PSRO. In contrast, Sparsification-PSRO maintains a compact population but fails to match Sparse PSRO's exploitability performance. This result demonstrates that the Sparsity Metric is critical for ensuring policy quality despite size constraints.

Together, these results confirm that the Sparsity Metric and Policy Space Sparsification act synergistically: the former guarantees the geometric distinctiveness of new policies via convex combination residual constraints, while the latter controls population size through thresholding. Their combination enables Sparse PSRO to balance exploration depth and computational efficiency, outperforming ablation variants in both equilibrium approximation and computational efficiency.

## 6 CONCLUSION

This paper presents Sparse PSRO to address policy redundancy in the processes of policy generation and policy population construction. Specifically, Sparse PSRO employs two innovations: Sparsity Metric for dissimilarity quantification and Policy Space Sparsification for geometrically distinct strategy admission. Theoretical analysis confirms finite policy population maintenance with guaranteed separation distances and the convergence to the Nash Equilibrium. Experimental validation across diverse environments further validates its superior performance in exploitability reduction and policy population compression.

**Limitation and Future Work.** Our current limitation is that the distance parameter in Policy Space Sparsification requires manual tuning. Future work will focus on developing adaptive thresholding mechanisms based on Policy Hull curvature analysis. In addition, we consider extending the convex combination residual metric to asymmetric game settings. We leave them as our future works.

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

## A  NOTATION

Notations are provided in Tab. 2.

Table 2: Notations

| Notation | Definition |
|---|---|
| $G$ | Symmetric zero-sum game with policy space $S$ and payoff matrix $U$ |
| $S$ | Set of pure strategies (deterministic actions) |
| $U$ | Payoff matrix satisfying $U = -U^\top$ (symmetric zero-sum) |
| $\pi$ | Individual pure strategy ($\pi \in S$) |
| $\pi_i$ | Mixed strategy (probabilistic distribution over $S$, $\pi_i \in \Delta(S)$) |
| $u$ | Mixed-strategy payoff: $u(\pi_1, \pi_2) = \pi_1 U \pi_2^\top$ |
| $(\pi_1^*, \pi_2^*)$ | Nash Equilibrium strategy profile |
| $\mathcal{E}(\pi)$ | Exploitability: deviation from Nash Equilibrium |
| $\Pi_i^t$ | Policy population at iteration $t$: $\{\pi_i^1, \ldots, \pi_i^t\}$ |
| $M$ | Meta-game payoff matrix for policy interactions |
| $\sigma_i$ | Meta-game Nash Equilibrium (mixed strategy over $\Pi_i^t$, $\sigma_i \in \Delta(\Pi_i^t)$) |
| $\pi_i^{t+1}$ | New best-response strategy against $\sigma_{-i}^t$ |
| $H(\Pi_i)$ | Policy Hull: convex hull of $\Pi_i$ |
| sparsity$(\cdot, \cdot)$ | Sparsity Metric: minimal squared distance to policy convex combinations |
| $\mathbf{c}$ | Convex combination combination coefficients for sparsity |
| $\lambda$ | Hyperparameter for exploration-exploitation balance |
| $\mu$ | Distance threshold in periodic update (add if $\geq \mu$ to $H(\Pi_i)$) |

# B   THEORETICAL ANALYSIS

## B.1   PROOF OF THEOREM 4.1

*Proof.* 1. In multi-agent games, mixed strategies are probability distributions over a finite set of pure strategies, forming a simplex $\Delta(S) \subset \mathbb{R}^{|S|}$, which is compact (closed and bounded) in Euclidean space. Let $\mathcal{S} = \Delta(S)$ denote this compact policy space.

2. By the threshold condition, a strategy is added to $\Pi_i^t$ only if sparsity$(\pi_i, \Pi_i^t) > \mu$. This means each newly added strategy $\pi_i$ lies outside the closed ball of radius $\sqrt{\mu}$ around the linear span of $\Pi_i^t$, ensuring geometric separation from existing strategies.

3. For any compact set $\mathcal{S}$, the $\epsilon$-packing number $P(\mathcal{S}, \epsilon)$, the maximum number of points in $\mathcal{S}$ where each pair is separated by at least $\epsilon$, is finite for all $\epsilon > 0$ (Anthony & Bartlett, 1999). Here, $\epsilon = \sqrt{\mu}$, as each new strategy $\pi_i$ must satisfy $\|\pi_i - \Pi_i^{t\top} \mathbf{c}\|_2 > \sqrt{\mu}$. Consequently, for any two policies $\pi_i^k, \pi_i^m \in \Pi_i^t$: $\|\pi_i^k - \pi_i^m\|_2 > \sqrt{\mu}$, ensuring a minimum separation distance in the policy vector space.

4. First, $\Pi_i^1$ is finite by definition. Second, suppose $\Pi_i^t$ is finite. When generating $\pi_i^{t+1}$, it is added to $\Pi_i^t$ only if it is $\sqrt{\mu}$-separated from the span of $\Pi_i^t$. Since $\mathcal{S}$ is compact, the packing number $P(\mathcal{S}, \sqrt{\mu})$ is finite, bounding the number of such separable strategies. Thus, $\Pi_i^{t+1}$ remains finite. By induction, $\Pi_i^t$ is finite for all $t$, and the policy population never grows exponentially. $\square$

## B.2   PROOF OF THEOREM 4.3

*Proof.* 1. By Theorem 3.1, the sparsification rule ensures any two strategies in $\Pi_i^t$ have a minimum separation of $\sqrt{\mu}$. Since the policy space is a compact subset of a finite-dimensional normed vector space, the size of $\Pi_i^t$ is bounded by the finite packing number, guaranteeing $\Pi_i^t$ remains finite for all $t$.

2. Suppose $\mathcal{PE}(\Pi^t) > 0$. There exists a best response $\pi_i^* = \text{BR}(\sigma_{-i}^t)$ not in $H(\Pi_i^t)$ (otherwise $\mathcal{PE}(\Pi^t) = 0$, contradicting the assumption). Since $\pi_i^* \notin H(\Pi_i^t)$, its sparsity sparsity$(\pi_i^*, \Pi_i^t) > 0$, and its utility $u(\pi_i^*, \sigma_{-i}^t)$ exceeds that of any strategy in $H(\Pi_i^t)$. For any strategy $\pi_i \in H(\Pi_i^t)$, its sparsity sparsity$(\pi_i, \Pi_i^t) = 0$, so its optimization objective value is $u(\pi_i, \sigma_{-i}^t)$. In contrast, $\pi_i^*$'s objective value is $u(\pi_i^*, \sigma_{-i}^t) + \lambda \cdot (> 0)$, which is strictly larger than that of $\pi_i$. Since $\pi_i^{t+1}$ is the optimal solution of this objective, it cannot lie in $H(\Pi_i^t)$, so $H(\Pi_i^t) \subsetneq H(\Pi_i^{t+1})$ (PH is strictly expanded).

3. Since the game is a symmetric zero-sum game, player $-i$ follows the same update logic as player $i$: if $\mathcal{PE}(\Pi^t) > 0$, its best response $\pi_{-i}^* \notin H(\Pi_{-i}^t)$, and the regularization-constrained optimal strategy $\pi_{-i}^{t+1}$ also lies outside $H(\Pi_{-i}^t)$, leading to $H(\Pi_{-i}^t) \subsetneq H(\Pi_{-i}^{t+1})$. According to the theoretical property of Policy Hull, expanding PH reduces population exploitability; thus, $\mathcal{PE}(\Pi^t) > \mathcal{PE}(\Pi^{t+1})$. Since PH expands strictly for both players, PE decreases monotonically:

$$\mathcal{PE}(\Pi^1) > \mathcal{PE}(\Pi^2) > \cdots > \mathcal{PE}(\Pi^t) \geq 0$$

4. By monotonicity and boundedness ($\mathcal{PE} \geq 0$), PE converges to a limit $\mathcal{PE}^* \geq 0$. Assume for contradiction that $\mathcal{PE}^* > 0$; then the global NE $\sigma^*$ is not in $H(\Pi_i^t) \times H(\Pi_{-i}^t)$ for any $t$. By Assumption 4.2, the threshold $\mu$ is sufficiently small to retain strategies critical for PH expansion, ensuring no key strategies for covering $\sigma^*$ are erroneously discarded. Given the finite policy population and strict PH expansion, $\sigma^*$ must eventually lie in $H(\Pi_i^t) \times H(\Pi_{-i}^t)$, contradicting $\mathcal{PE}^* > 0$.

Thus, $\mathcal{PE}^* = 0$, and the policy population contains a global NE. $\square$

# C   ALGORITHM FOR TWO ABLATION VARIANTS

## C.1   SPARSIFICATION-PSRO

Pseudo-code of Sparsification-PSRO is given in Algorithm 2. Sparsification-PSRO removes the convex combination residual constraint, retaining only the best-response optimization objective to isolate the effect of sparsity regularization.

---

**Algorithm 2** Sparsification-PSRO

---

**Input** Initial policy populations $\Pi_i^1, \Pi_{-i}^1$

1: Compute payoff matrix $M_{\Pi_i^1, \Pi_{-i}^1}$

2: Initialize meta policies $\sigma_i^1 \sim \text{UNIFORM}(\Pi_i^1)$

3: **for** $t = 1, 2, \ldots$ **do**

4:     **for** player $i \in \{1, 2\}$ **do**

5:         Initialize $\pi_i = \pi_i^t$

6:         Sample $K$ policies $\{\pi_i^k\}_{k=1}^K$ from Policy Hull $\Pi_i^t$

7:         **for** many episodes **do**

8:             Sample $\pi_{-i} \sim \sigma_{-i}^t$     ▷ Sample opponent policy from meta-policy distribution

9:             Update $\pi_i$ over $\mathcal{BR}(\sigma_{-i}^t)$   ▷ Update policy via standard best-response optimization

10:         **end for**

11:         $\pi_i^{t+1} = \pi_i$

12:         **if** sparsity$(\pi_i, \Pi_i^t) > \mu$ **then**         ▷ Threshold-controlled policy admission

13:             $\Pi_i^{t+1} = \Pi_i^t \cup \{\pi_i\}$     ▷ Admit only geometrically distinct policies to population

14:         **end if**

15:     **end for**

16:     Compute missing entries in $M_{\Pi_i^t, \Pi_{-i}^t}$

17:     Compute meta-strategies $(\sigma_i^{t+1}, \sigma_{-i}^{t+1})$ from $M_{\Pi_i^t, \Pi_{-i}^t}$

18: **end for**

**Output** current meta-strategy for each player.

---

## C.2   SPARSITY-PSRO

---

**Algorithm 3** Sparsity-PSRO

---

**Input** Initial policy populations $\Pi_i^1, \Pi_{-i}^1$

1: Compute payoff matrix $M_{\Pi_i^1, \Pi_{-i}^1}$

2: Initialize meta policies $\sigma_i^1 \sim \text{UNIFORM}(\Pi_i^1)$

3: **for** $t = 1, 2, \ldots$ **do**

4:     **for** player $i \in \{1, 2\}$ **do**

5:         Initialize $\pi_i = \pi_i^t$

6:         Sample $K$ policies $\{\pi_i^k\}_{k=1}^K$ from Policy Hull $\Pi_i^t$

7:         **for** many episodes **do**

8:             Sample $\pi_{-i} \sim \sigma_{-i}^t$     ▷ Sample opponent policy from meta-policy distribution

9:             Update $\pi_i$ over Eq. 7     ▷ Apply sparsity-regularized best-response optimization

10:         **end for**

11:         $\pi_i^{t+1} = \pi_i$

12:         $\Pi_i^{t+1} = \Pi_i^t \cup \{\pi_i\}$     ▷ Expand policy population without screening

13:     **end for**

14:     Compute missing entries in $M_{\Pi_i^t, \Pi_{-i}^t}$

15:     Compute meta-strategies $(\sigma_i^{t+1}, \sigma_{-i}^{t+1})$ from $M_{\Pi_i^t, \Pi_{-i}^t}$

16: **end for**

**Output** current meta-strategy for each player.

---

Pseudo-code of Sparsity-PSRO is given in Algorithm 3. Sparsity-PSRO replaces the threshold-controlled phase with a continuous dense exploration mode to verify the role of the strategy addition rule in exploration-exploitation balance.

# D   EXPERIMENT DETAILS

## D.1   EXPERIMENTAL ENVIRONMENTS

The detailed description of each experimental environment is as follows:

- Random Game of Skills (RGoS): A synthetic environment with configurable strategy interactions, enabling controlled validation of exploration dynamics.

- AlphaStar: A high-complexity StarCraft II meta-game, challenging the algorithm in real-world, high-dimensional policy spaces.

- Blotto: A continuous-space resource allocation game, demanding efficient exploration-exploitation balance.

- Kuhn Poker: An incomplete-information card game, evaluating robustness under uncertainty and partial observability.

More experimental results are presented in Fig. 5 and Fig. 6, where Sparse PSRO exhibits consistent superiority across all extended environments.

## D.2 EXPERIMENTAL SETUP

All experiments follow a unified evaluation framework to ensure result comparability and conclusion reliability. All experimental results are based on the mean ± standard deviation of 10 independent runs with random seeds to ensure statistical reliability. Error bars in the figures uniformly represent one standard deviation from the mean. Experiments are run on a personal computer with a 16-core CPU and 16GB of memory. Each single experimental run for each environment takes 4–6 hours to complete. Through systematic hyperparameter tuning, this study determined that the sparsity weight $\lambda = 0.1$ and the distance threshold $\mu = 0.02$. Relevant code and configuration files will be publicly released alongside the paper to ensure reproducibility.

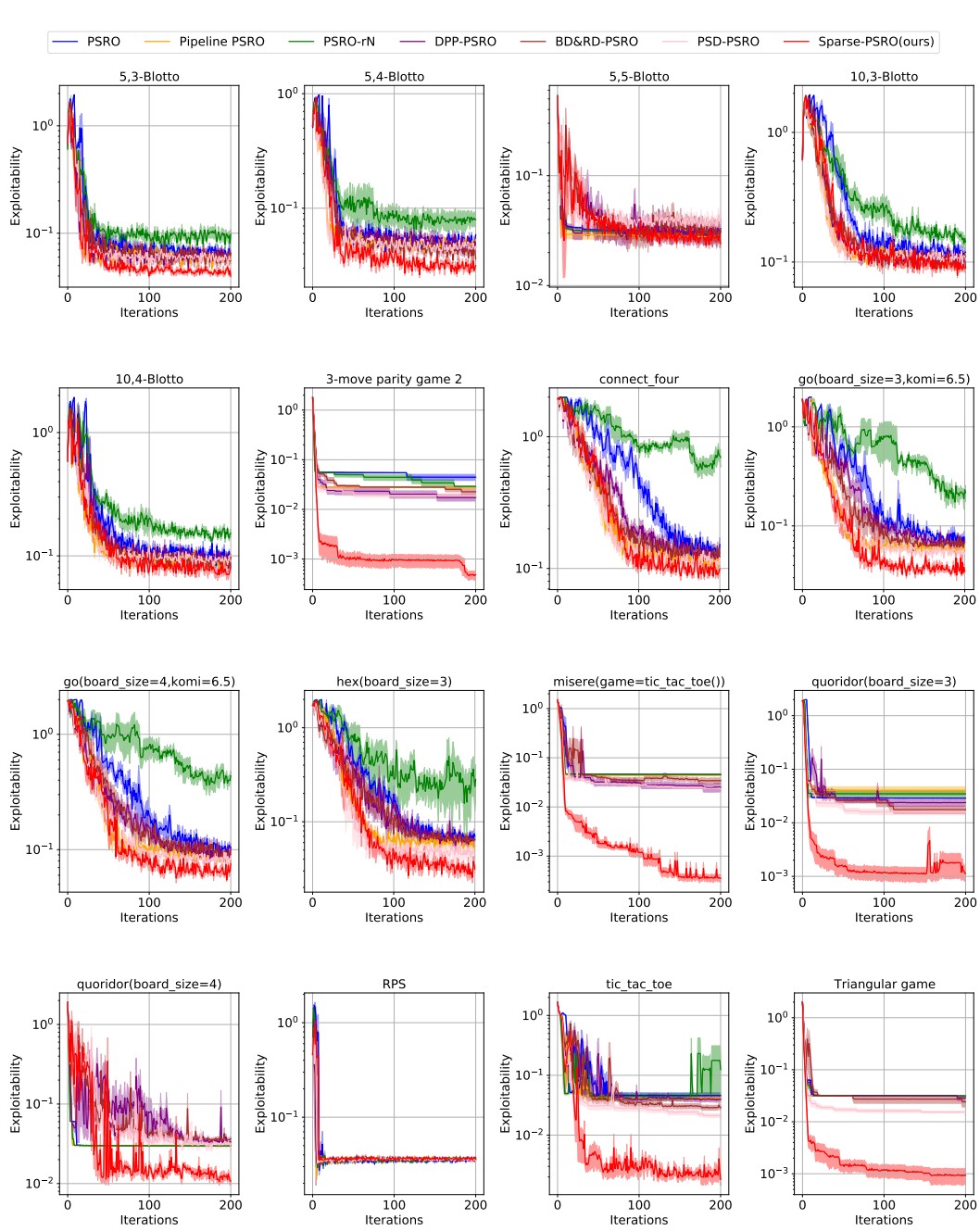

Figure 5: Comparison of exploitability across more environments

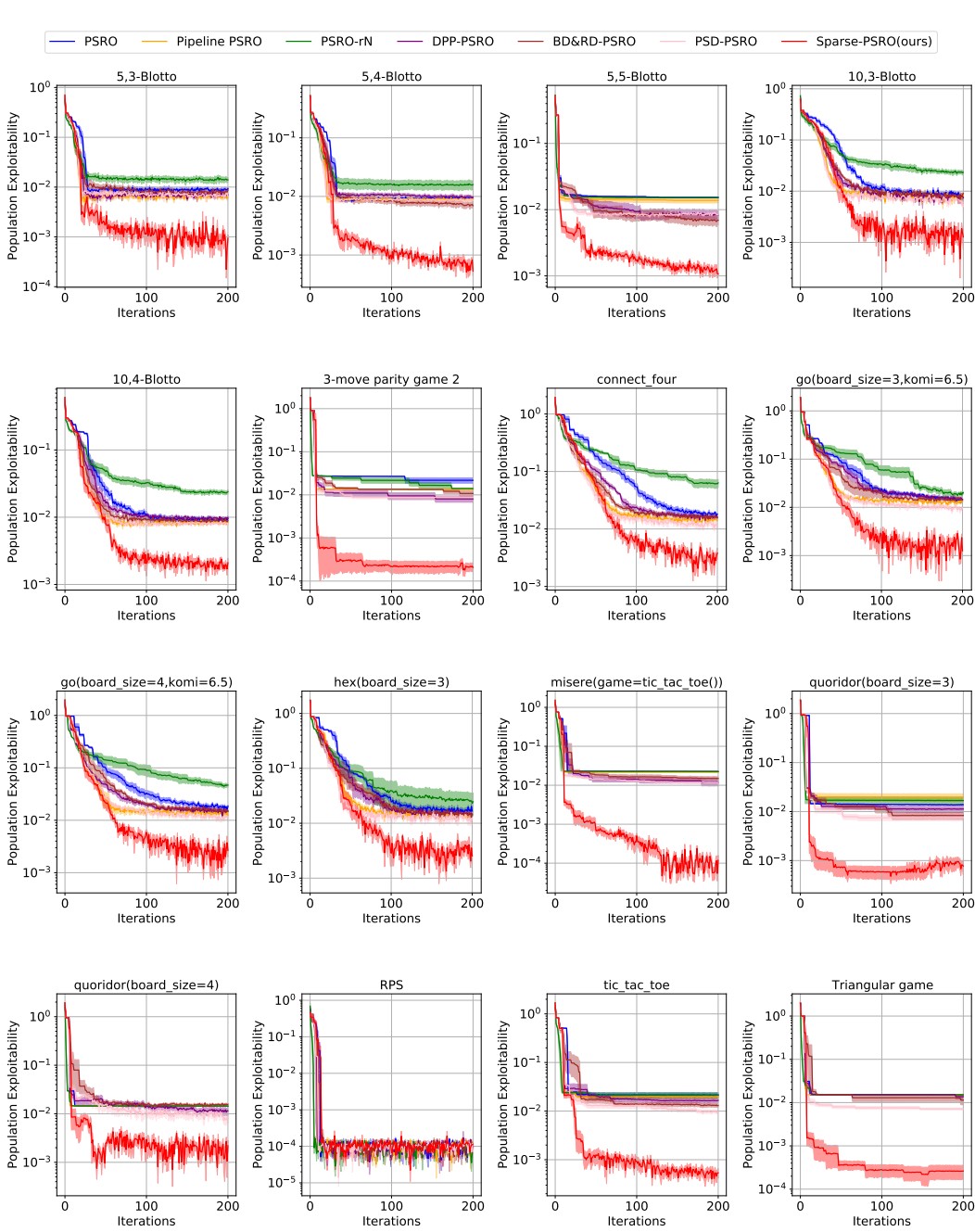

Figure 6: Comparison of population exploitability across more environments

