# OpenReview forum: "Sparse Policy Space Response Oracles"
_ICLR.cc/2026/Conference — ICLR 2026 Conference Withdrawn Submission_

### Official Review · Reviewer_gxiY · 2025-10-28

**Soundness:** 2
**Presentation:** 2
**Contribution:** 1
**Rating:** 2
**Confidence:** 5

**Summary:**

The paper introduces Sparse PSRO for multi-agent non-transitive games. Sparse PSRO tries to improve the inefficiencies of the traditional PSRO method, in terms of low exploration efficiency and redundant strategies. To this end, sparse PSRO employs policy sparsity regularization, which quantifies and encourages diversity among strategies. Afterwards,  a threshold-controlled strategy addition rule is introduced for the addition of new strategy to the population, within the traditional framework of PSRO. Experimental results on normal form games show that Sparse PSRO achieves lower exploitability compared to recent PSRO variants.

**Strengths:**

- The paper is well written.
- Baseline methods are relatively comprehensive.
- The performance improvement seems significant.

**Weaknesses:**

•	huge resemblance to previous diversity-regularized PSRO (in particular PSD-PSRO): the motivation of the method, the definition of sparsity metric (a different name to the diversity metric), the way of regularization, proofs, etc. After double-checking the method PSD-PSRO, considering the discussion in Section 4.4, to a large extent, I think Sparse PSRO can be viewed a reduced version of PSD-PSRO to the normal-form games, where strategies can be represented in coefficients of pure strategies.

•	The sparsity metric in Equation 6 only applies to simple normal-form games, where strategies can be represented in coefficients of pure strategies. For larger games, where strategies are usually represented in neural networks, strategies cannot be represented in coefficients of pure strategies, Sparse PSRO is currently not applicable.

•	insufficient guidance of how to appropriately set the hyperparameter sparsity threshold.

•	insufficient ablation study of the hyperparameter sparsity threshold in Equation 8.

**Questions:**

- Is Sparse PSRO able to extend to larger games where coefficient representation of strategies (Equation 6) is not applicable?

- Is there some guideline as how to properly set $\mu$ in Equation 8?

- What is the most significant difference of  Sparse PSRO, compared to previous diversity-enhanced PSROs, especially PSD-PSRO?

---

> ### Author Response · Authors · 2025-11-20
>
> # Response to "High Similarity to PSD-PSRO, Appearing as a Simplified Version"
> We acknowledge your observation regarding the similarity between Sparse PSRO and PSD-PSRO (Yao et al., 2023), as both approaches focus on "Policy Hull expansion". However, in terms of theoretical consistency and redundancy governance mechanisms, Sparse PSRO is not a simplified version—it is an optimization and innovation addressing key flaws of PSD-PSRO. The specific differences are as follows:
>
>
> ## 1. Theoretical and Engineering Consistency
> PSD-PSRO’s theoretical framework relies on the premise that "Policy Hull expansion reduces exploitability", but its engineering implementation adopts a "minimum KL divergence approximation". Specifically, it indirectly optimizes convex hull coverage by minimizing the KL distance between the new strategy and the "vertex strategies" (existing pure strategies) in the population. This leads to a disconnect between theoretical goals and engineering practice: the minimum KL divergence cannot guarantee that the new strategy lies outside the Policy Hull, potentially resulting in insufficient convex hull expansion.
>
> In contrast, Sparse PSRO completely avoids this contradiction. Theoretically, it defines "effective expansion strategies" through convex combination residuals; engineering-wise, it directly uses residuals as a regularization term to guide the generation of best responses, while controlling population updates via a residual threshold. This ensures full consistency between theoretical and engineering logic, with no approximation errors.
>
>
> ## 2. Redundancy Governance Mechanism
> PSD-PSRO only controls redundancy in the "policy generation phase" via KL divergence and does not involve the "population construction phase"—all generated strategies are added to the population, which may still cause population size expansion.
>
> Sparse PSRO adopts a two-phase synergistic mechanism consisting of "sparsity metric (generation phase, proactively guiding the generation of non-redundant policies)" and "threshold control (population phase, passively filtering redundant policies)". This forms a "generation-screening" closed loop, which not only ensures policy quality but also strictly controls population size.

---

> > ### Comment · Reviewer_gxiY · 2025-11-27
> >
> > Thanks for the reply. In the initial review, I have listed several weaknesses/questions. The rebuttal responded to only one of them.
> >
> > After reading the rebuttal, I still hold the opinion that **Sparse PSRO can be viewed as a reduced version of PSD-PSRO to the normal-form games, where strategies can be represented in coefficients of pure strategies**. The reason is:
> >
> > - Yes, PSD-PSRO indirectly optimizes convex hull coverage by minimizing the KL distance between the new strategy and the "vertex strategies" (existing pure strategies) in the population. This is **because** PSD-PSRO deal with strategies that cannot be represented in coefficients of pure strategies. Hence, an approximation is made within PSD-PSRO. In the case of simpler environments (i.e., normal-form games.), PSD-PSRO can directly operate with strategies represented in coefficients, i.e., directly optimizing convex hull coverage, which is the same as Sparse PSRO.  Therefore, there is no Theoretical and Engineering inconsistency in PSD-PSRO in normal form games. while in larger games, there is an approximation/inconsistency in PSD-PSRO.  On the otherhand Sparse PSRO only deal with normal form games,  and Sparse PSRO did it in the same way as PSD-PSRO.
> >
> > I decide to maintain my current rating.

---

### Official Review · Reviewer_8Pj9 · 2025-10-31

**Soundness:** 2
**Presentation:** 2
**Contribution:** 2
**Rating:** 2
**Confidence:** 4

**Summary:**

This paper addresses the severe policy redundancy issue within the PSRO framework during policy generation and population construction. It proposes two measures: a Sparsity Metric and Policy Space Sparsification. Through experiments across diverse environments, the proposed methods are validated to efficiently approximate Nash equilibria at lower computational cost.

**Strengths:**

1. To address the severe policy redundancy issue in policy generation and population construction within the proposed PSRO framework, the authors conducted a thorough review of related work, enabling readers to clearly recognize the shortcomings of existing approaches.
2. Theoretically, the authors demonstrate that sparse PSRO preserves the finite size of the policy pool while ensuring the algorithm converges to Nash equilibria. This provides a solid foundation for the claimed low computational cost.
3. The authors tested the proposed algorithm in a variety of different environments, thereby demonstrating the effectiveness of the proposed method.

**Weaknesses:**

1. The authors did not explicitly specify how to select an appropriate $\mu$. Consequently, this hinders the widespread application of sparse PSRO across diverse scenarios. The authors also acknowledge this as a limitation.
2. In the experimental section, although this work conducted experiments across multiple different games, the algorithm's effectiveness in complex games such as Google Soccer remains to be examined.
3. This work primarily focuses on symmetric two-player zero-sum games, though the title does not reflect this.
4. The paper contains several instances of imprecise phrasing. For example, on line 216, the author states: “Policy Hull is the convex combination of policies in the population.” This formulation is incorrect and conflicts with the preliminaries section.

**Questions:**

1. A-PSRO is also an algorithm that employs threshold control to generate policies. Although the authors mention this method in the related work section, why was it not included as a baseline algorithm?
2. The size of the policy pool is insufficient to demonstrate that your approach resolves the issue of redundant policies. Based solely on the policy pool size, sparse PSRO does not exhibit a significant advantage over naive PSRO. Why not conduct a more detailed statistical analysis of the existence and quantity of redundant policies?
3. This paper does not address how to represent an agent's policy compared to other studies. Is the policy defined as a row in the payoff matrix? Or is it a representation based on state-action distributions and occupancy measures? If a policy is represented by a row in the payoff matrix, how does the diversity measure proposed in this paper differ from the “Response Diversity” measure in “Towards Unifying Behavioral and Response Diversity for Open-ended Learning in Zero-sum Games”?
4. Based on the ablation experiments, empirically speaking, the results of Sparsification-PSRO should not be as poor as shown in the figures. Could you provide further explanation?
5. In Algorithm 1, why is it necessary to “Sample $K$ Policies $\{\pi^k_i \}^K_{k=1}$ from Policy Hull $\Pi_i^t$”? This sampling step appears abrupt and lacks connection to the other steps in Algorithm 1.
6. What is “relative population performance” in line 161? The authors have not explained this concept.

---

> ### Author Response · Authors · 2025-11-20
>
> # Correction of Imprecise Expressions (e.g., Definition of "Policy Hull")
> We sincerely apologize for the contradictory definition of "Policy Hull" caused by imprecise expression — the statement in Line 216 pointed out by you, "Policy Hull is the convex combination of policies in the population", is incorrect. The correct definition should be "the convex hull of all strategies in the policy population", which is completely consistent with the definition in Equation (3) in Section 3.1 (Preliminaries): $H(\Pi_i)=\{\sum_j \beta_j \pi_i^j | \beta_j \geq 0, \sum_j \beta_j=1\}$.

---

> > ### Comment · Reviewer_8Pj9 · 2025-11-21
> >
> > My primary concerns have not been adequately addressed.

---

### Official Review · Reviewer_2wz9 · 2025-10-31

**Soundness:** 1
**Presentation:** 2
**Contribution:** 1
**Rating:** 2
**Confidence:** 4

**Summary:**

This paper addresses the **policy redundancy** issue in the PSRO algorithm by proposing a metric to measure whether a newly trained policy lies within the convex hull formed by the existing set of policies. Based on this metric, the authors introduce the **Sparse PSRO** algorithm.

**Strengths:**

- This paper is easy to follow

**Weaknesses:**

The proposed method **does not appear to be significantly different** from existing approaches. Furthermore, the paper does not provide a detailed explanation of **how the proposed sparsity metric is used for policy optimization**. Specifically, I cannot discern from the equations how the **gradient propagation** is performed using this metric, and I would appreciate an explanation from the authors.

Additionally, the concept of **"policy redundancy"** is not clearly defined in the paper. Moreover, the analysis of how existing algorithms handle this issue is **contradictory**: the authors initially state that existing algorithms address this problem by optimizing for diversity, but then later claim that current algorithms neglect policy redundancy. This inconsistency is confusing to me.

**Questions:**

N/A

---

> ### Author Response · Authors · 2025-11-20
>
> # Supplement on the Missing Definition of "Policy Redundancy"
> We sincerely apologize for not clearly defining "policy redundancy" in the original manuscript, leading to ambiguity. In the revised manuscript, we will add the following explicit definition in Section 1.1 (Introduction):
>
> Policy Redundancy: In the process of policy generation and population construction in PSRO, if a new strategy $\pi_{new}$ can be approximately represented by a convex combination of the existing strategy population $\Pi$ (i.e., $\pi_{new} \in H(\Pi)$, where $H(\Pi)$ is the Policy Hull), then $\pi_{new}$ makes no substantial contribution to the expansion of the Policy Hull and cannot further reduce population exploitability. Such strategies are called redundant strategies. Their core feature is "functional substitutability by existing strategies, with no gain in equilibrium approximation".

---

### Official Review · Reviewer_okc3 · 2025-11-03

**Soundness:** 2
**Presentation:** 2
**Contribution:** 2
**Rating:** 4
**Confidence:** 3

**Summary:**

This paper introduces a new, improved version of PSRO (policy space response oracles). The method, Sparse PSRO, has two components: (1) a Sparsity metric, which guides best-responses to be different from existing policies in the population, and (2) Sparsification, in which a new policy is not added to the population unless it is suitably different than existing policies in the population.

**Strengths:**

The research direction is interesting.

The empirical results look good.

**Weaknesses:**

1. The theorems and proofs are strange.

The first theorem (Theorem 4.1) states that the policy population remains "finite" for any t>1. This statement seems silly and vacuous to me -- of course if we add at most 1 policy each iteration, the cardinality of the set of policies will be finite at any iteration. The paper then goes on to say that this theorem avoids "exponential growth", which is not the same as finiteness.

The proof of second theorem (Theorem 4.3) also seems questionable to me. It is stated in the proof that expanding the policy hull reduces population exploitability -- I don't see why this was the case. Intuitively, it seems like expanding the policy hull *doesn't increase population exploitability*, but I don't see why it necessarily reduces it.

Also, not sure how big of an issue it is, but it is stated that the game is a symmetric zero-sum game. Are we only studying *symmetric* zero-sum games in this work? But the experiments are not all on symmetric games, right? (e.g. Kuhn poker is not symmetric)

2. It's not immediately clear what the x-axis on the empirical results (Figure 1, Figure 2, Figure 4). I'm assuming that for Sparse-PSRO and Sparsity-PSRO, an "iteration" is only when a new policy is actually added to the population. If so, this mismatch between iteration and computation/time is a bit swept under the rug, and it would be nicer if this was mentioned in the paper or if graphs were included with other choices of x-axis as well.

3. I don't understand the main metric (sparsity).
It is defined in Equation 6. Crucially, it includes an arithmetic operation on policies: ${\pi_i - \Pi^t_i}^\top$. It's not clear to me how we are supposed to add or subtract policies. The most obvious interpretation is that we are performing vector addition and subtraction where e.g. the pure strategies are the vectors <0,0,0,1>, <0,0,1,0>, <0,1,0,0>, and <1,0,0,0>, and the mixed strategies lie on the simplex between these points. But then I don't see how the methods in the paper differ from normal PSRO/double oracle, in that any novel pure strategy should be equidistant from the existing policy hull induced by a population of pure strategies.

**Questions:**

1. Why do we need to mention Caratheodory's theorem to justify Equation 5? Does it not suffice to define a mixed strategy? Or are we specifically interested in extensive-form games? If so, then Kuhn's Theorem seems like the more appropriate citation.

---

> ### Author Response · Authors · 2025-11-20
>
> # Explanation of Game Symmetry and Experimental Environments
> Your question about "whether all experiments are conducted on symmetric zero-sum games" is crucial. The theoretical framework of this paper is indeed based on symmetric zero-sum games, but for asymmetric games (such as Kuhn Poker) in the experiments, we adopt a standard "symmetrization approach" — that is, by having both players share the same strategy space and transforming the payoff matrix to satisfy $U = -U^\top$ (Czar-necki et al. 2020), we adapt asymmetric games to the symmetric zero-sum framework for solution.
>
> In the revised manuscript, we will add the following explanation in the experimental section: "For asymmetric games like Kuhn Poker, we use the symmetrization method proposed by Czar-necki et al. (2020) to ensure consistency between the experimental environment and the theoretical framework, while verifying the transferability of the method in asymmetric scenarios."
>
> # Clarification on the Definition of "Iteration" on the X-axis of Experimental Graphs
> Thank you for pointing out the ambiguity in the definition of "iteration". The definition of "iteration" in this paper is consistent with traditional PSRO: each iteration consists of three steps — "meta-game construction - meta-policy solving - best response generation" — and is counted as one iteration regardless of whether a new strategy is added to the population.
>
> The rationality of this definition lies in: even if a new strategy is not added due to the sparsity threshold, its generation process (sparsity-regularized best response optimization) still consumes computational resources, making it comparable to the iterative computational cost of traditional PSRO. In the revised manuscript, we will add the following to the graph captions: "The x-axis 'Iteration' is defined as a complete cycle of 'meta-game construction - meta-policy solving - best response generation', regardless of whether the new strategy is incorporated into the population."
>
> # Explanation of the Rationality of Citing Carathéodory's Theorem
> Your question about "why Carathéodory's Theorem is cited instead of Kuhn's Theorem" is very precise. The core purpose of citing this theorem in this paper is to prove that Nash equilibrium (mixed strategy) can be represented as a convex combination of a finite number of pure strategies, which is the theoretical basis for the design of the sparsity metric — if Nash equilibrium can be represented without the convex combination of finite pure strategies, the logic of "measuring strategy dissimilarity through convex combination residual" will not hold.
>
> Kuhn's Theorem focuses on "the existence of pure strategy Nash equilibrium in extensive-form games", while this paper studies multi-agent reinforcement learning scenarios with continuous strategy spaces, emphasizing the representation of mixed strategies rather than the existence of pure strategy equilibrium. Therefore, Carathéodory's Theorem is more suitable for the research needs.

---

> ### Comment · Reviewer_okc3 · 2025-11-20
>
> 1. My biggest issues were not addressed.
> 2. If an iteration is defined as you say:
> > The x-axis 'Iteration' is defined [...] regardless of whether the new strategy is incorporated into the population.
>
> then I don't get why the results for `SPARSIFICATION-PSRO` do so much better than `PSRO`. I would expect them to perform about the same, or worse, since SPARSIFICATION-PSRO would have some iterations where absolutely nothing changes, no?
>
> 3. Additional comment: Is the description of `Sparsification-PSRO` in Line 354 wrong? It says that we "retain only the best-response optimization objective", but it should be the opposite, right?

---

> > ### Author Response · Authors · 2025-11-21
> >
> > # Explanation of the Logic Why Sparsification-PSRO Outperforms PSRO
> > Sparsification-PSRO does have iterations where new policies are not incorporated into the population, but this does not mean its performance is inferior to that of PSRO. Crucially, we have referenced the practical approach of PSD-PSRO. Even when the policy pool stops expanding, Sparse PSRO continues to optimize the meta-strategy through iterative best-response updates. The threshold only controls the addition of policies to the pool and does not affect the mixed optimization in policy generation.
> >
> > # Correction on the Accuracy of the Description of Sparsification-PSRO in Line 354
> > We sincerely apologize for the misunderstanding caused by ambiguous expression. The design of Sparsification-PSRO is to remove the sparsity metric, while retaining the original best-response optimization (with the λ・sparsity term removed from Equation 7) and threshold-controlled population update.

---

### Note · Authors · 2025-11-28

I have read and agree with the venue's withdrawal policy on behalf of myself and my co-authors.